The first dromaeosaurid (Dinosauria: Theropoda) from the Lower Cretaceous Bayan Gobi Formation of Nei Mongol, China

Pittman Michael 1 mpittman@hku.hk
Pei Rui 1
Tan Qingwei 2
Xu Xing 3
1 Vertebrate Palaeontology Laboratory, Department of Earth Sciences, The University of Hong Kong , Pokfulam , Hong Kong, China
2 Long Hao Institute of Geology and Paleontology , Hohhot, Nei Mongol , China
3 Key Laboratory of Vertebrate Evolution and Human Origins, Institute of Vertebrate Paleontology & Paleoanthropology, Chinese Academy of Sciences , Beijing , China
Edwards Scott
Electronic publication date: 2015 Dec 7
Publication date: 2015
Volume: 3
Electronic Location ID: e1480
Received 2015 Sep 1; Accepted 2015 Nov 16
Copyright: © 2015 Pittman et al.
Copyright year: 2015
Copyright holder: Pittman et al.
License: This is an open access article distributed under the terms of the Creative Commons Attribution License, which permits unrestricted use, distribution, reproduction and adaptation in any medium and for any purpose provided that it is properly attributed. For attribution, the original author(s), title, publication source (PeerJ) and either DOI or URL of the article must be cited.
License URL: https://creativecommons.org/licenses/by/4.0/

Keywords: Deinonychosaur, Dromaeosaurid, Paravian, Theropod, Microraptorine, Bayan Gobi Formation, Nei Mongol, Balongwula, Lower Cretaceous

Funding: National Science Foundation of China 41120124002 Research Grant Council’s General Research Fund 17103315 The collection and study of IVPP V22530 was supported by funds awarded to XX by the National Science Foundation of China (41120124002) and the Bureau of Land and Resources of Bayan Nur. Study of IVPP V22530 was also supported by funds awarded to MP by the Research Grant Council’s General Research Fund (17103315). The funders had no role in study design, data collection and analysis, decision to publish, or preparation of the manuscript.

==============================
The first dromaeosaurid theropod from the Lower Cretaceous Bayan Gobi Formation is identified based on an incompletely preserved partially-articulated left leg, increasing the known diversity of its understudied ecosystem. The leg belongs to specimen IVPP V22530 and includes a typical deinonychosaurian pedal phalanx II-2 with a distinct constriction between the enlarged proximal end and the distal condyle as well as a typical deinonychosaurian enlarged pedal phalanx II-3. It possesses a symmetric metatarsus and a slender and long MT V that together suggest it is a dromaeosaurid. Two anatomical traits suggest the leg is microraptorine-like, but a more precise taxonomic referral was not possible: metatarsals II, III and IV are closely appressed distally and the ventral margin of the medial ligament pit of phalanx II-2 is close to the centre of the rounded distal condyle. This taxonomic status invites future efforts to discover additional specimens at the study locality because—whether it is a microraptorine or a close relative—this animal is expected to make important contributions to our understanding of dromaeosaurid evolution and biology. IVPP V22530 also comprises of an isolated dromaeosaurid manual ungual, a proximal portion of a right theropod anterior dorsal rib and an indeterminate bone mass that includes a collection of ribs. Neither the rib fragment nor the bone mass can be confidently referred to Dromaeosauridae, although they may very well belong to the same individual to whom the left leg belongs.

Introduction

Dromaeosaurid theropod dinosaurs are rare to find, but their close evolutionary relationship to birds make them important subjects in studies of avian origins and powered flight (Turner, Makovicky & Norell, 2012). They are currently known from the Cretaceous of both Laurasia (modern Asia, North America, and Europe) and Gondwana (modern South America and Africa [and potentially Antarctica]) (Turner, Makovicky & Norell, 2012), but are inferred to have Jurassic origins (Hu et al., 2009). With only ∼30 known taxa (Turner, Makovicky & Norell, 2012), every new dromaeosaurid specimen is potentially a valuable one. This study describes a specimen from a new dromaeosaurid locality in Inner Mongolia, China: Balongwula (巴隆乌拉) of the Bayan Gobi Formation (Fig. 1). The Bayan Gobi Formation preserves a Lower Cretaceous terrestrial environment that includes three dinosaurs described by Sino-Canadian, Sino-Belgian and American teams: the psittacosaurid ornithischian Psittacosaurus gobiensis (Russell & Zhao, 1996; Sereno, 2010), the hadrosauroid ornithischian Penelopognathus weishampeli (Godefroit, Li & Shang, 2005) and the therizinosauroid theropod Alxasaurus elesitaiensis (Russell & Dong, 1994). The formation also preserves fish, bivalves, gastropods, ostracods and plants (BGMRNMAR, 1991; Russell & Dong, 1994; Zhou & Dean, 1996). Here we broaden the dinosaurian diversity of this formation and extend the distribution of dromaeosaurids northwest of existing Chinese Lower Cretaceous localities (Zhou, Barrett & Hilton, 2003) as well as southeast of existing Mongolian ones (Turner, Hwang & Norell, 2007).

Materials and Methods

IVPP V22530, an incomplete partially-articulated left dromaeosaurid leg (Fig. 2; distal portion of the femur (Fig. 3), tibia and fibula (Fig. 3), MTs I-V (Fig. 4) and pedal phalanges II-1, II-2, II-3, III-1, III-3?, IV-1 and IV-2? (Fig. 4)), an isolated dromaeosaurid manual ungual (Fig. 5), a proximal portion of a right theropod anterior dorsal rib (Fig. 6) and an indeterminate bone mass including a collection of ribs (Fig. 7). It is uncertain how the leg is related to the other skeletal elements, but it is possible that they might all belong to the same individual.

IVPP V22530 was collected, studied and described using standard palaeontological methods, in accordance with a fossil excavation permit (13-07-ELT) obtained from the Department of Land and Resources, Nei Mongol, China.

Locality and Horizon

IVPP V22530 was discovered in Balongwula (巴隆乌拉), Nei Mongol, China (∼41.0°N, 105.8°E; ∼130 km Northwest of the city of Bayan Nur (巴彦淖尔市, ∼40.8°N, 107.4°E) by a field team from the Institute of Vertebrate Paleontology and Paleoanthropology (IVPP), Beijing. The locality is part of the Lower Cretaceous (Aptian (Jerzykiewicz & Russell, 1991; Russell & Dong, 1994) to Albian (Kielan-Jaworowska, Cifelli & Lu, 2005)) Bayan Gobi Formation and comprises of claystones, mudstones, siltstones and limestones that appear to have all been deposited in a terrestrial environment (BGMRNMAR, 1991; Zhou & Dean, 1996). IVPP V22530 itself is preserved in a dark grey ostracod mudstone, possibly deposited in a lake or pond.

Figure 1 Location of Balongwula (巴隆乌拉), Nei Mongol, China where IVPP V22530 was discovered.

Scale = 200 km.

Description and Comparisons

Left hindlimb

The main portion of IVPP V22530 comprises of an incomplete partially-articulated left dromaeosaurid leg preserved between a main slab and counterslab (Fig. 2).

Figure 2 IVPP V22530 includes an incomplete, partially-articulated left dromaeosaurid leg.

(A) main slab; (B) counterslab. Scale = 2 cm.

The distal portion of the femur in IVPP V22530 appears to be curved (Fig. 3) suggesting that the entire bone was presumably bowed, as in most theropods. An ectocondylar tuber is present on the posterior edge of the lateral condyle (Figs. 3C and 3D; Brusatte et al., 2014: Character 411 state 0), a feature common to theropod dinosaurs. The tibiotarsus is slender (length/mid-shaft width ratio ∼ 18; Fig. 2B), comparable to those of basal dromaeosaurids, but differs from the robust tibiotarsus of derived dromaeosaurids such as Deinonychus (ratio ∼ 13, Ostrom, 1969: Table 10), Velociraptor (ratio ∼ 10, Norell & Makovicky, 1999: Fig. 10C) and Linheraptor (ratio ∼ 11, Xu et al., 2010: Fig. 1). The tibial shaft is preserved in anteromedial view and is medially concave (Figs. 2B, 3A and 3B), as in most theropods including Velociraptor (MPC 100/986, Norell & Makovicky, 1999: Figs. 11A and 11C), Microraptor (CAGS 20-8-001, Hwang et al., 2002: Fig. 28A) and the oviraptorosaur Gigantoraptor (LH V0011, Xu et al., 2007: Fig. 1S). The distal end of the tibiotarsus is missing so nothing can be said about the morphology of the astragalus and calcaneum. However, the tibiotarsus is estimated to be ∼7 cm long. The proximal end of the fibula is expanded and proximal to its mid-shaft there is an iliofibularis tubercle (Figs. 3A and 3B; Brusatte et al., 2014: Character 820 state 0) that is anterolaterally projecting, as in other theropods (Rauhut, 2003: Character 211 state 1; Fig. 48). The latter projects in a similar direction to Mahakala (MPC 100/1033, Turner, Pol & Norell, 2011: Figs. 32A and 32C), unlike in Rahonavis and ornithurines where this tubercle faces posteriorly (Forster et al., 1998: Fig. 4C). The distal end of the fibula is missing, but it probably had a complete length of less than 7 cm.

Figure 3 Femur, tibia and fibula of IVPP V22530.

(A) Photograph, and (B) line drawing of the partial distal portion of the femur and the proximal half of the tibia and fibula from the counterslab of the left leg. Scale = 2 cm. (C) Photograph, and (D) line drawing of a close-up of the partial distal portion of the femur and the proximal end of the tibia and fibula from the main slab of the left leg. Scale = 1 cm.

Figure 4 Left metatarsus and foot of IVPP V22530.

(A) Photograph, and (B) line drawing of counterslab. Scale = 1 cm. (C) photograph, and (D) line drawing of main slab. Scale = 1 cm.

The metatarsus is slender (Fig. 4) and like most other dromaeosaurids it is symmetric (Brusatte et al., 2014: Character 205 state 0). In contrast, troodontids have an asymmetrical metatarsus as metatarsal (MT) II is more slender than MT IV (Brusatte et al., 2014: Character 205 state 1). The metatarsus is ∼60% of the length of the tibiotarsus (∼4 cm and ∼7 cm long respectively), but a more accurate ratio is unavailable as the proximal portion of the metatarsus and the distal portion of the tibiotarsus are missing from the specimen (Fig. 2). MT I is a reduced and broadly triangular bone that is attached to the medial side of the distal end of MT II (Figs. 4C and 4D), as in most theropods (Rauhut, 2003: Character 222, state 2), including troodontids (e.g., Mei long [IVPP 12733, Xu & Norell, 2004: Fig. 1B] and basal birds (e.g., Archaeopteryx [JM 2257, Wellnhofer, 2009: Fig. 5.85). However, in Microraptor MT I is attached medioventrally to the distal end of MT II (Hwang et al., 2002: Fig. 30A; Pei et al., 2014: Fig. 15). MT II, III and IV are closely appressed distally (Figs. 4C and 4D) as in microraptorines and Buitreraptor (MPCA 245, Makovicky, Apesteguía & Agnolin, 2005: Fig. 3I), whereas in Mahakala (MPC 100/1033, Turner, Pol & Norell, 2011: Figs. 33A and 33C), Rahonavis (UA 8656, Turner, Makovicky & Norell, 2012: Fig. 56A), Velociraptor (MPC 100/985, Norell & Makovicky, 1997: Fig. 7) and Deinonychus (YPM 5205, Ostrom, 1969: Fig. 73) the distal end of the metatarsus is not appressed. This intraclade variation is also seen in ornithomimids (e.g., appressed in Sinornithomimus [IVPP V11797-23, Kobayashi & Lü, 2003: Fig. 23A]; unappressed in Qiupalong [HGM 41HIII-0106, Xu et al., 2011: Fig. 2D]) and alvarezsauroids (e.g., appressed in Linhenykus [IVPP 17608, Xu et al., 2013: Fig. 12A1; unappressed in Alvarezsaurus [MUCPv 54, Chiappe, Norell & Clark, 2002: Fig. 4.25B]), but to our knowledge the distal ends of the metatarsals in troodontids are always appressed (Makovicky & Norell, 2004: Fig. 9.6). The seemingly ginglymus distal end of MT II is insufficiently preserved to confirm the related dromaeosaurid synapomorphy: a ginglymus distal end that extends onto the extensor surface and gives the distal end a strongly concave profile in anterior view (Brusatte et al., 2014: Character 198 state 1). The distal ends of MT II and IV extend as far as each other (Figs. 4C and 4D; Brusatte et al., 2014: Character 433 state 0) as in several dromaeosaurids, including Buitreraptor (MPCA 245, Makovicky, Apesteguía & Agnolin, 2005: Fig. 3I), Zhenyuanlong (JPM 0008, Lü & Brusatte, 2015: Fig. 3B) and Sinornithosaurus (IVPP V12811, Xu, Wang & Wu, 1999: Fig. 2). However, in some dromaeosaurids MT II and IV have different lengths as in Microraptor (Hwang et al., 2002: Fig. 30A), Graciliraptor (IVPP V13474, Xu & Wang, 2004: Fig. 3E), Deinonychus (YPM 5205, Ostrom, 1969: Table 11, Fig. 71; AMNH 3015, Ostrom, 1969: Table 11), Velociraptor (MPC 100/985, Norell & Makovicky, 1997: Fig. 7) and Adasaurus (MPC 100/20, Turner, Makovicky & Norell, 2012: Fig. 13A). In troodontids, MT II is also shorter than MT IV (Makovicky & Norell, 2004: Fig. 9.6). MT III is only slightly longer than MT II (Figs. 4C and 4D) like that of Sinornithosaurus (IVPP V12811, Xu, Wang & Wu, 1999: Fig. 4F) and Buitreraptor (MPCA 245, Makovicky, Apesteguía & Agnolin, 2005: Fig. 3I), whereas many other dromaeosaurids show a large difference in length including Graciliraptor (IVPP V13474, Xu & Wang, 2004: Fig. 3E), Microraptor (Hwang et al., 2002: Fig. 30A) and Rahonavis (UA 8656, Forster et al., 1998: Fig. 4E). A distinct flange (ridge) is developed along the (latero)ventral edge of MT IV (Figs. 4C and 4D; Brusatte et al., 2014: Character 226 state 1) which is restricted to troodontids and some dromaeosaurids such as Neuquenraptor (MCF PVPH 77, Novas & Pol, 2005: Fig. 1F), Buitreraptor (MPCA 245, Makovicky, Apesteguía & Agnolin, 2005), Bambiraptor (AMNH 30554, Burnham et al., 2000), Velociraptor (MPC 100/986, Norell & Makovicky, 1999: Fig. 16B) and Microraptorinae (e.g., Microraptor [BMNHC PH881, Pei et al., 2014: Fig. 15]; Sinornithosaurus [IVPP V12811, Xu, Wang & Wu, 1999: Fig. 1]; Changyuraptor [HG B016, Han et al., 2014: Fig. 2]). In IVPP V22530 (Figs. 4C and 4D), this flange is less developed than in Neuquenraptor (MCF PVPH 77, Novas & Pol, 2005: Fig. 1F), Microraptor (BMNHC PH881, Pei et al., 2014: Fig. 15) and Sinornithosaurus (IVPP V12811, Xu, Wang & Wu, 1999: Fig. 4F). MT V is slender and elongate—approximately half the length of MT IV (Figs. 4C and 4D)—as in Microraptorinae (e.g., Microraptor [BMNHC PH881, Pei et al., 2014: Fig. 15]; Sinornithosaurus, [IVPP V12811, Xu, Wang & Wu, 1999: Fig. 1]; Changyuraptor [HG B016, Han et al., 2014: Fig. 2]) and many other dromaeosaurids (e.g., Deinonychus [YPM 5205, Ostrom, 1969: Fig. 73]; Velociraptor [MPC 100/985, Norell & Makovicky, 1997: Fig. 7]; Bambiraptor [AMNH FR 30556, Burnham, 2003: Table 3.4]). The only dromaeosaurid we know of with a comparatively short MT V is Balaur (EME PV 313, Brusatte et al., 2013: Figs. 36 and 37) which is around a third of the length of MT IV (but the distal tip of the right MT V is broken [EME PV 313, Brusatte et al., 2013: Fig. 37C]), although a recent studies argue that it is actually an avialan (Cau, Brougham & Naish, 2015; Foth, Tischlinger & Rauhut, 2014; Godefroit et al., 2013). Thus, a long MT V is potentially a dromaeosaurid synapomorphy. The preserved distal portion of MT V is broken along the lateral plane making it difficult to determine its cross-sectional shape (Figs. 4C and 4D). However, its lateral edges taper distally (Figs. 4C and 4D), like in other theropods (Rauhut, 2003) including Microraptor (LVH 0026, Gong et al., 2012: Fig. 7), Velociraptor (MPC 100/985, Norell & Makovicky, 1997: Fig. 7) and Deinonychus (YPM 5205, Ostrom, 1969: Fig. 73).

Pedal phalanx II-2 has a typical deinonychosaurian profile with a distinct constriction between the enlarged proximal end and the distal condyle (Figs. 4C and 4D; Brusatte et al., 2014: Character 201 state 1). Saurornitholestes and Microraptorinae are atypical in this regard as the constriction is less developed (Longrich & Currie, 2009: Figs. 2B and 2D). The oval-shaped, dorsally-offset medial ligament pit of phalanx II-2 is deep and its ventral margin is close to the centre of the rounded distal condyle, as in most microraptorines, Saurornitholestes (Longrich & Currie, 2009: Fig. 2B2) and Bambiraptor (AMNH 30554, Burnham et al., 2000). In Rahonavis (UA 8656, Forster et al., 1998: Fig. 4D), Neuquenraptor (MCF PVPH 77, Novas & Pol, 2005: Fig. 1H), Deinonychus (YPM 5205, Ostrom, 1969: Fig. 74), Dromaeosaurus (AMNH FARB 5356, Turner, Makovicky & Norell, 2012: Fig. 40A), Adasaurus (MPC 100/21, Barsbold, 1983) and Velociraptor (MPC 100/985, Norell & Makovicky, 1997: Fig. 6B) the pit is similar, but its ventral margin is more dorsally-positioned. The microraptorine Hesperonychus (TMP 1983.67.7, Longrich & Currie, 2009: Fig. 2B2) has a similar condition to the latter taxa, but its pit is more circular-shaped. A proximodorsal lip is developed on phalanx II-3 (Figs. 4C and 4D; Brusatte et al., 2014: Character 731 state 0), as in most theropods. This phalanx is enlarged as with all deinonychosaurians (Brusatte et al., 2014: Character 201 state 1). Phalanx III-1 is straight and longer than phalanges II-1 and IV-1 (Fig. 4), as in most theropods.

Manual ungual

IVPP V22530 also includes an isolated theropod manual ungual (Fig. 5) which was found in association with the incomplete partially-articulated left dromaeosaurid leg. Although ungual morphology is often undiagnostic, there is evidence, albeit limited, suggesting that these fossils belong to the same individual. The manual ungual measures ∼16 mm from the proximodorsal corner to the tip, matching the leg in size if they are referable to Microraptorinae. The small articular surface, the well-developed flexor tubercle and a gracile, heavily-curved profile, in combination with a transverse ridge present immediately dorsal to the articular surface (‘proximodorsal lip’; Brusatte et al., 2014: Character 150 state 1), suggest that this manual ungual is referable to Pennaraptora (Foth, Tischlinger & Rauhut, 2014). Furthermore, the ungual is dorsally arched (the dorsal margin is above the proximal end when the proximal articular facet is positioned vertically) as in Dromaeosauridae (Senter et al., 2004), though not as strongly arched as in the latter, supporting the dromaeosaurid affinity of the specimen. The ventrodistal portion of the ungual comprises of a vertical keel that is probably a fossilised keratinous sheath.

Figure 5 An isolated theropod manual ungual is associated with the incomplete partially-articulated left dromaeosaurid leg.

Scale = 1.6 cm.

Anterior dorsal rib

A triangular proximal end of a single right rib and the proximal portion of its shaft are preserved (Fig. 6). It is not clear if it belongs to the same dromaeosaurid whose left leg is preserved. The capitulum and tuberculum are subequal in size, but the former is more dorsally-positioned than the latter and is supported by a distinct neck that separates it from the rest of the rib’s proximal proportion (Fig. 6). The relative length of the rib shaft is unknown because its distal portion is missing, but the degree of tapering present in the shaft suggests that it is probably relatively short (Fig. 6A). The tuberculum, the dorsal and lateral margins of the rib roughly form a right-angle, whilst the capitulum makes a ∼45° angle with the lateral margin (Fig. 6A). This combination of features resembles the anterior dorsal ribs of Deinonychus (YPM 5245, Ostrom, 1969: Fig. 51A; YPM 5204, YPM 5210, Schachner et al., 2011: Fig. 4J) and Microraptor (CAGS 20-7-004, Hwang et al., 2002: Figs. 15B and 15C). This suggested position along the spine is supported by the presence of reasonably strong forking between the tuberculum and capitulum because the degree of forking decreases along the ribcage of theropods (e.g., Allosaurus [Schachner et al., 2011: Fig. 4H], Tyrannosaurus [FMNH PR2081, Schachner et al., 2011: Fig. 4I] and Deinonychus [YPM 5204, YPM 5210, Schachner et al., 2011: Fig. 4J]). Dromaeosaurid anterior dorsal ribs possess few taxonomically-informative characteristics and unfortunately it is not possible to confirm if there is a deep groove along the anterior edge of the rib shaft, as observed in Microraptor (IVPP V12330, Xu, 2002; CAGS 20-8-001, Hwang et al., 2002: Fig. 13; BMNHC PH881, Pei et al., 2014: Fig. 6), because the surface of the proximal portion of the shaft is damaged and because—as mentioned—the distal half is missing. Of note is a rectangular-shaped process on the anterior surface of the rib that is located ventromedial to the tuberculum (Fig. 6A). The anterior dorsal ribs of Mahakala (IGM 100/1033, Turner, Pol & Norell, 2011: Fig. 15) and Microraptor (CAGS 20-7-004, Hwang et al., 2002: Fig. 15B) differ from the rib of IVPP V22530 in having a dorsoventrally lower and mediolaterally wider tuberculum and a smaller capitulum on the tip of longer and thinner neck. In Velociraptor the anterior dorsal ribs (MPC-D100/54, Hone et al., 2012: Fig. 1) differ from the rib of IVPP V22530 in having a long and thin neck that ends in a smaller capitulum. There is no rectangular-shaped process amongst the anterior dorsal ribs of Mahakala, but this is uncertain for the other taxa mentioned. Thus, IVPP V22530 is probably an anterior dorsal rib, although its exact position along the vertebral series (rib number) is unknown.

Figure 6 IVPP V22530 includes an isolated proximal portion of a right dorsal rib.

Scale = 5 mm.

Indeterminate bone mass

The elements that make up IVPP V22530 also include an indeterminate mass of bones that include some tiny ribs (Fig. 7). This bony mass was found in close proximity to the other elements, but does not contain any recognisable dromaeosaurid bones. These bones might be non-dinosaurian so have been given to relevant experts for further identification.

Figure 7 An indeterminate bone mass that includes some rib-like elements was found in the vicinity of the leg, ungual and rib elements that make up IVPP V22530.

Scale = 1 cm.

Discussion

Taxonomic status

Pedal phalanx II-2 has a typical deinonychosaurian profile with a distinct constriction between the enlarged proximal end and the distal condyle (Brusatte et al., 2014: Character 201 state 1). Pedal phalanx II-3 is enlarged as in deinonychosaurians (Brusatte et al., 2014: Character 201 state 1). The symmetry of the metatarsus of IVPP V22530 (Brusatte et al., 2014: Character 205 state 0) indicates that it is a dromaeosaurid (Brusatte et al., 2014: Character 205 state 0) rather than a troodontid, as the latter displays an assymetrical metatarsus (Brusatte et al., 2014: Character 205 state 1), though some microraptorine dromaeosaurids also have a somewhat assymetrical metatarsus (e.g., Sinornithosaurus [IVPP V12811, Xu & Wang, 2000: Fig. 2, Table 1]). The presence of a slender and long MT V suggests that IVPP V22530 is probably a dromaeosaurid. The distal end of MT II appears to be ginglymus, but it is insufficiently preserved to confirm the related dromaeosaurid synapomorphy (Brusatte et al., 2014: Character 198 state 1). A dorsally arched manual ungual with a small articular facet and a large flexor tubercle also suggests the dromaeosaurid affinity of the specimen. IVPP V22530 is therefore a dromaeosaurid theropod.

MT II, III and IV are closely appressed distally (Fig. 4B) as in microraptorines and Buitreraptor (MPCA 245, Makovicky, Apesteguía & Agnolin, 2005: Fig. 3I), whereas in other dromaeosaurids (Ostrom, 1969; Norell & Makovicky, 1997; Turner, Pol & Norell, 2011; Turner, Makovicky & Norell, 2012) the distal end of the metatarsus is not appressed. As with most microraptorines, Saurornitholestes and Bambiraptor the oval-shaped, dorsally-offset medial ligament pit of phalanx II-2 is deep in IVPP V22520 and its ventral margin is close to the centre of the rounded distal condyle. However, IVPP V22530 differs from Microraptorinae and Saurornitholestes in having a more developed constriction between the enlarged proximal end and the distal condyle of pedal phalanx II-2, as in other dromaeosaurids (Longrich & Currie, 2009: Figs. 2B and 2D; Brusatte et al., 2014: Character 201 state 1). Thus, the metatarsus of IVPP V22530 suggests that the individual has affinites with microraptorines and Buitreraptor whilst the position of the medial ligament pit of phalanx II-2 supports the former and suggests similarities with Saurornitholestes and Bambiraptor. However, the relative constriction between the enlarged proximal end and the distal condyle of this phalanx contradicts the specimen’s affinities with microraptorines and Saurornitholestes. It is unclear at present what the relative weighting of these three anatomical traits should be in assessing the specimen’s taxonomic status, but from a functional perspective the more direct weight-bearing role of the metatarsus should produce more conservative individual variations in its bones in comparison to the pedal phalanges. From this perspective the metatarsus might be a more reliable source of taxonomically-informative information, but this hypothesis needs to be confirmed through further study. Thus, with the evidence available, IVPP V22530 is tenatively referred to Microraptorinae or its close relatives (Fig. 8). A phylogenetic analysis is beyond the scope of this study, but IVPP V22530 will be included in a future analysis of paravian interrelationships.

Figure 8 A palaeoreconstruction of IVPP V22530 next to its inferred depositional setting, a muddy lake environment.

© Julius T. Csotonyi.

Body size estimation

Femoral circumference has been shown to be a more faithful proxy of theropod body mass and by measuring this parameter in Mahakala it was estimated that this animal weighed less than a kilogram (0.79 [0.59–0.98]) (Campione et al., 2014). IVPP V22530 is not sufficiently well-preserved to measure femoral circumference, but its shorter tibiotarsus length (∼75 mm) compared to Mahakala (left tibial length = 110.0 mm; Turner, Pol & Norell, 2011: Table 1) suggests that IVPP V22530 weighed less than Mahakala (<1 kg) as tibiotarsus length is also correlated with body mass, albeit in a weaker way (Campione et al., 2014: Table 2).

Ontogenetic stage

In the absence of histological data, an individual’s ontogenetic stage can potentially be constrained using the surface texture (Tumarkin-Deratzian & Vann, 2006; Tumarkin-Deratzian, Vann & Dodson, 2007) and fusion (Brochu, 1995; Brochu, 1996; Irmis, 2007) of its bones. In IVPP V22530, only the bone surfaces of the fibula and pedal phalanges II-2 and III-1 are exposed (Figs. 2, 3A, 3C and 4C). However, their quality of preservation is insufficient to characterise their texture for the purpose of assessing ontogenetic stage. Fusion between the tarsals themselves and with the tibia or the metatarsus can indicate an adult/subadult ontogenetic stage for an individual, as in Balaur (Brusatte et al., 2013). However, if these bones are unfused this does not necessarily mean that the individual is a juvenile (e.g., Mahakala (Turner et al., 2007b; Turner, Pol & Norell, 2011) and Bambiraptor (Burnham et al., 2000)). The ankle region of IVPP V22530 is not preserved on either slab and the surfaces of the limb bones are either damaged or unexposed (Fig. 2). Thus, the ontogenetic stage of this specimen is uncertain as histological analysis was not possible to conduct owing to the poor preservation of the tibia and fibula (Figs. 2 and 3).

Palaeoenvironmental inferences

The Bayan Gobi Formation preserves a diverse terrestrial vertebrate fauna that includes mammals, champsosaurs, trionychids and other turtles (Russell & Dong, 1994), as well as the ornithischian dinosaurs Psittacosaurus gobiensis (Russell & Zhao, 1996; Sereno, 2010) and Penelopognathus weishampeli (Godefroit, Li & Shang, 2005) and the theropod dinosaur Alxasaurus elesitaiensis (Russell & Dong, 1994). It also preserves a range of flowering and non-flowering plants (BGMRNMAR, 1991; Russell & Dong, 1994; Zhou & Dean, 1996) with the conifers Classopollis and Podocarpites (Russell & Dong, 1994) potentially indicating a cooler palaeoclimate. Fish, bivalve, gastropod (including Viviparus) and ostracod (including Cypridea) fossils (BGMRNMAR, 1991; Russell & Dong, 1994; Zhou & Dean, 1996) indicate the presence of lakes, ponds or rivers in the original ecosystem because there were no nearby oceans at this time (Scotese, 2001). This is consistent with the numerous suspected ostracod carapaces in the matrix of IVPP V22530 (i.e., they look like ostracod carapaces based on their outline, but there is no further morphological information such as hingement and muscle scars preserved or observed; Fig. 3). The mudstone matrix of IVPP V22530 could also have been deposited in those depositional settings, but the abundance of carbonaceous plant fossil fragments in the rock unit as well as its dark colour—that is presumably related to its high-organic content—suggests that a relatively high trophic index lake might be a better candidate instead. It is important to note that detailed sedimentological correlations across Bayan Gobi Formation fossil sites are still wanting and many of the specimens mentioned have yet to be described in detail, thus, the picture of the palaeoecosytem painted above remains a tentative one.

Conclusions

IVPP V22530 comprises of an incompletely preserved partially-articulated left dromaeosaurid leg, an isolated dromaeosaurid manual ungual, a proximal portion of a right theropod dorsal rib and an indeterminate bone mass that includes a collection of ribs. Two anatomical traits suggest that the left leg belongs to a microraptorine or a close relative: metatarsals II, III and IV are closely appressed distally and the ventral margin of the medial ligament pit of phalanx II-2 is close to the centre of the rounded distal condyle. This referral means that IVPP V22530 is the first described dromaeosaurid—and small-sized theropod (<1 kg)—from the Bayan Gobi Formation, helping to expand our understanding of this understudied Lower Cretaceous ecosystem. Aptian to Albian ages have been specifically suggested for the formation, but constraining them further would be invaluable as a well-supported Albian age could make IVPP V22530 the first-known Albian microraptorine-like dromaeosaurid. As a microraptorine IVPP V22530 would extend the geographical range of this clade because the study site is ∼500 km northwest of Liaoning Province, which is the only area where Lower Cretaceous microraptorines are known. It would also fill a temporal gap between the Barremian/Aptian-aged microraptorines and the Campanian-aged microraptorine Hesperonychus (Longrich & Currie, 2009). As a close microraptorine relative IVPP V22530 would be the first non-North American example. Thus, further discoveries at the study site will help fill important gaps in our knowledge of dromaeosaurid evolution and biology between the Aptian/Albian and Campanian stages of the Cretaceous.

Supplemental Information

Supplemental Information 1 Chinese language abstract

A Chinese language abstract for interested locals from Nei Mongol and other parts of China.

Click here for additional data file.

We would like to thank Ding Xiaoqin for preparing the specimen as well as Moriaki Yasuhara for discussing the ostracods of IVPP V22530 with MP.

Additional Information and Declarations

Competing Interests

Author Contributions

Field Study Permissions

Data Availability

The authors declare there are no competing interests.

Michael Pittman and Xing Xu conceived and designed the experiments, performed the experiments, analyzed the data, contributed reagents/materials/analysis tools, wrote the paper, prepared figures and/or tables, reviewed drafts of the paper.

Rui Pei performed the experiments, analyzed the data, wrote the paper, prepared figures and/or tables, reviewed drafts of the paper.

Qingwei Tan conceived and designed the experiments, performed the experiments, analyzed the data, contributed reagents/materials/analysis tools, prepared figures and/or tables, reviewed drafts of the paper.

The following information was supplied relating to field study approvals (i.e., approving body and any reference numbers):

IVPP V22530 was collected, studied and described using standard palaeontological methods, in accordance with a fossil excavation permit (13-07-ELT) obtained from the Department of Land and Resources, Nei Mongol, China.

The following information was supplied regarding data availability:

The research in this article did not generate any raw data—all data collected is presented in the manuscript.

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
