# Peer review of "The first dromaeosaurid (Dinosauria: Theropoda) from the Lower Cretaceous Bayan Gobi Formation of Nei Mongol, China"

_PeerJ, doi:10.7717/peerj.1480_

## Round 0.1 · original submission · Minor Revisions

I want to apologize for the long time this article took to review. Most of that time was due to one reviewer who thought his/her review was no longer needed. At any rate, please be aware that this is unusual for this journal and I'm sorry it took so long in your case.

As you can see both reviewers recommended minor revision, and were generally supportive of publication. However, reviewer 1 had some critiques on the quality of the figures and the location data. This reviewer also felt that you should perform a phylogenetic analysis to confirm the proposed taxonomic hypothesis. The phylogenetic analysis may be beyond the scope of the paper but I encourage you to at least mention the need for one if you don't actually do one.

Reviewer 2 had mostly minor suggestions but also felt that the figures needed improvement. Please do your best to improve the quality of the figures and add any additional figures recommended.

Reviewer 1 ·

Basic reporting

The manuscript from Pittman et al. entitled “The first dromaeosaurid (Dinosauria: Theropoda) from the Early Cretaceous Bayan Gobi Formation of Nei Mongol, China” describes new fragmentary theropod remains from Nei Mongol, which probably belongs to a dromaeosaurid. The paper is generally well written and I propose minor revisions. However, there are still some aspects that need to be addressed before the paper can be published.

•The introduction is very brief. I recommend extending this section, presenting some more information, e.g. what other vertebrate remains are known from the formation? Who had worked in this area before? Maybe some general information on dromaeosaurids and their biogeography.

•Fig. 1. This is a really nice panorama photo, but not very informative. I propose that the authors include a drawing of a map of this area, showing the locality of the material and the extension of the Bayan Gobi Formation. If available the authors may also include a stratigraphic profile of the formation. Furthermore, I am a little bit confused: The locality shown in Fig.1 is different from the GPS coordinates and the location of the village Elesitai mentioned in the text. According to Google maps the GPS coordinates show a locality, which is about 120 km northwest from Bayan Nur (close to the locality of Penelopognathus, see Fig. 2 in Godefroit et al. 2005), while the village Elesitai is c. 35 km southwest to Alxa (in the area were Alxasaurus was found). I am not an expert in the geography of China, but something seems to be wrong. Could the authors please explain these contradictions and correct the error in the text and/or figure.

•L94ff: The authors may add that the distal ends of the tibiotarsus and the fibula are missing and that nothing can be said about the morphology of the astragalus and calcaneus. What is the estimated length of these bones?

•L98f: How is the situation in Deinonychus?

•L107ff: Just a question. Is it possible to judge the projection of the iliofibularis tubercle on the basis of the preservation of the material?

•L115ff: What is the estimated length of the metatarsus?

•L122f: You can also cite Pei et al. 2014 with respect to the attachment of MT I. How is the situation in other dromaeosaurids, troodontids and basal Avialae?

•L123ff: How is the situation in alvarezsauroids? Are the distal ends of the MTs appressed to each other or not?

•L138ff: This situation (MT II shorter than MT IV) is also present in troodontids (see Makovicky & Norell 2004) and Deinonychus (see Ostrom 1969).

•L142ff: Somehow this statement is similar to that in L116f. If the authors prefer to keep this information in, they should present it together with the description in L116f.

•L153: Citation for the situation in Microraptorinae.

•L158: Citation for the situation in dromaeosaurids.

•L162: The authors may also cite Godefroit et al. (2013) and Foth et al. (2014), who also found Balaur nesting within Avialae.

•L185: Change from “Phalanx III-1 is straight and is longer than phalanges II-1 and IV-1” to “Phalanx III-1 is straight and longer than phalanges II-1 and IV-1”.

•L214ff: Concerning the rib head morphology: How is this morphology depending on the rib position (see Schachner et al. 2011)? How is the situation in other non-dromaeosaurid theropods?

•L223f: Could the authors mark the rectangular process in the Fig. 4a?

•L240ff: No other type of bone can be identified?

•Discussion: Based on the material presented I have little doubt that the hind limb remains belong to a dromaeosaurid. However, the authors may consider testing their statement that the specimen represents a Microraptorinae with help of a phylogenetic analysis.

•L209ff: The authors may also take Field et al. (2013) for body mass estimation into account. According to this study the body mass (in birds) correlates significantly with the femur and metatarsus diameter. As both parameters should be possible to estimate from the material, the authors could test their body mass prediction directly.

•L30ff: The authors may cite Brochu (1995, 1996, but see also Irmis 2007) for bone fusion and Tumarkin-Deratzian et al. (2006, but also 2007) for bone surface as proxy for the ontogenetic stage, while Tumarkin-Deratzian (2009), Chiappe & Göhlich (2010), Dal Sasso & Salgado (2011) and Rauhut et al. (2012) as practical examples for dinosaur fossils. The authors may rephrase the initial sentence of the paragraph into the subjunctive form, because of Irmis (2007) and Tumarkin-Deratzian et al. (2007). Furthermore, I would appreciate if the authors could describe the bone surface of the specimen to confirm their observation regarding the ontogenetic stage of the specimen on the basis of the tarsal fusion.

•I further propose that the authors improve the quality of the figures. This is one of the few crucial issues in the manuscript. The photos are often way too dark to evaluate any anatomical details.

References:
Brochu CA. 1995. Heterochrony in the crocodylian scapulocoracoid. Journal of Herpetology 29:464–468.

Brochu CA. 1996. Closure of neurocentral sutures during crocodilian ontogeny: implications for maturity assessment in fossil archosaurs. Journal of Vertebrate Paleontology 16:49–62.

Chiappe LM, Göhlich UB. 2010. Anatomy of Juravenator starki (Theropoda: Coelurosauria) from the Late Jurassic of Germany. Neues Jahrbuch für Geologie und Paläontologie, Abhandlungen 258:257–296.

Dal Sasso C, Maganuco S. 2011. Scipionyx samniticus (Theropoda: Compsognathidae) from the Lower Cretaceous of Italy. Memorie della Società Italiana di Scienze Naturali e del Museo Civico di Storia Naturale di Milano 37:1–281.

Field DJ, Lynner C, Brown C, Darroch SAF. 2013. Skeletal correlates for body mass estimation in modern and fossil flying birds. PLoS ONE 8:e82000.

Godefroit P, Cau A, Hu D, Escuillié F, Wu W, Dyke GJ. 2013. A Jurassic avialan dinosaur from China resolves the early phylogenetic history of birds. Nature 498:359–362.

Irmis RB. 2007. Axial skeleton ontogeny in the Parasuchia (Archosauria: Pseudosuchia) and its implications for ontogenetic determination in archosaurs. Journal of Vertebrate Paleontology 27:350–361.

Makovicky PJ, Norell MA. 2004. Troodontidae. In: Weishampel DB, Dodson P, Osmólska H eds. The Dinosauria. Berkeley: University of California Press, 184–195.

Rauhut OWM, Foth C, Tischlinger H, Norell MA. 2012. Exceptionally preserved juvenile megalosauroid theropod dinosaur with filamentous integument from the Late Jurassic of Germany. Proceedings of the National Academy of Sciences 109:11746–11751.

Schachner ER, Farmer CG, McDonald AT, Dodson P. 2011. Evolution of the dinosauriform respiratory apparatus: new evidence from the postcranial axial skeleton. The Anatomical Record 294:1532–1547.

Tumarkin-Deratzian AR, Vann DR, Dodson P. 2006. Bone surface texture as an ontogenetic indicator in long bones of the Canada goose Branta canadensis (Anseriformes: Anatidae). Zoological Journal of the Linnean Society 148:133–168.

Tumarkin-Deratzian AR, Vann DR, Dodson P. 2007. Growth and textural ageing in long bones of the American alligator Alligator mississippiensis (Crocodylia: Alligatoridae). Zoological Journal of the Linnean Society 150:1–39.

Tumarkin-Deratzian AR. 2009. Evaluation of long bone surface textures as ontogenetic indicators in centrosaurine ceratopsids. The Anatomical Record 292:1485–1500.

Experimental design

No Comments

Validity of the findings

Please confirm the taxonomic classification with help of a phylogenetic analysis and increase the quality of the figures.

Reviewer 2 ·

Basic reporting

No Comments

Experimental design

No Comments

Validity of the findings

No Comments

Additional comments

Please see attached pdf for specific in-text comments.
In addition to these suggestions, I think it would be useful to have an additional figure that is a close up of the metatarsal and pes in Figure 1A. An interpretive line drawing would be a useful addition as well. The same is true for Figure 4 as well. An interpretive line drawing would be useful as a reader.

The figure of the pedal ungual is mislabeled as figure 3 when it should be figure 5. This image should be cropped down to just the ungual since most of the image is just matrix.

The thoracic rib figure is labeled as figure 4 when it should be figure 6. This image should be full page width.

The indeterminate bone mass figure is labeled as figure 5 when it should be figure 7.

The reconstruction image is labeled as figure 6 when it should be figure 8.

Annotated reviews are not available for download in order to protect the identity of reviewers who chose to remain anonymous.

---

## Round 0.2 · accepted · Accept

Thank you for attending to the figure quality. Your abstract in Chinese is especially appreciated and sets a good precedent.